# Risk factors for the prognosis of patients with sepsis in intensive care units

**Xiaowei Gai[1], Yanan Wang[2], Dan Gao[2], Jia Ma[3], Caijuan Zhang[4], Qiuyan Wang [1] ***

1 Department of Intensive Care Unit, Qinhuangdao Jungong Hospital, Qinhuangdao, Hebei, P. R. China,
2 Department of Operating Room, Qinhuangdao Jungong Hospital, Qinhuangdao, Hebei, P. R. China,
3 Department of General Surgery, Qinhuangdao Jungong Hospital, Qinhuangdao, Hebei, P. R. China,
4 Department of Anesthesiology, Tangshan Gongren Hospital, Tangshan, Hebei, P. R. China

* zichen0215@126.com

## Abstract

### Background and purpose

To date, sepsis remains the main cause of mortality in intensive care units (ICU). This study aimed analyze the risk factors of the prognosis in sepsis patients.

### Methods

In this case-control study, patients with sepsis admitted to the intensive care unit of a Chinese hospital between January and November 2020 were analyzed. Ultrasound and clinical data were analyzed and compared between non-survivors and survivors. The ROC curve analysis was also performed to determine the best indicator for predicting mortality.

### Results

A total of 72 patients with sepsis in ICU were included for analysis. The basic characteristics between the survivals and non-survivals were similar, except for acute physiology and chronic health evaluation (APACHE) II score, sepsis-related organ failure assessment (SOFA) score, lactate level, ultrasound parameters from superior mesenteric artery (SMA) such as peak systolic velocity (PSV), end-diastolic velocity (EDV) and resistive index (RI). Univariate analysis revealed that the APACHE II score, SOFA score, lactate, low PSV, EDV, and RI were potential risk factors for mortality in sepsis, while multivariate analysis suggested that low PSV was an independent risk factor for mortality, and the adjusted odds ratio was 0.295 (95% CI: 0.094–0.925). The ROC analysis showed that the PSV (AUC = 0.99; sensitivity and specificity were 0.99 and 0.96, respectively) had good predictive value for mortality in sepsis.

### Conclusion

Low PSV as found to be an independent risk factor and good predictor for mortality in patients with sepsis. This study shows the promise of ultrasound in predicting mortality in patients with sepsis; however, further studies are needed to validate these results.

**Data Availability Statement:** All relevant data are within the manuscript and its Supporting Information files.

**Funding:** This study was supported by the Medical Science Research Project of Health Commission of

Hebei Province, China [20201308].The funders had no role in study design, data collection and analysis, decision to publish, or preparation of the manuscript.

**Competing interests:** The authors have declared that no competing interests exist.

## Introduction

Hitherto, sepsis remains the main cause of mortality in intensive care units (ICUs) [1, 2], and shock accounts for one of the highest rates of mortality, approximately 50%, in many countries [3]. Therefore, early detection and management is of chief importance. The gastrointestinal tract has long been hypothesized to play an integral role in the occurrence and development of sepsis [4]. During septic shock, the body responds by redistributing blood to vital organs, such as the heart, brain, and kidneys. However, the gastrointestinal tract is the first organ to be affected by hypoperfusion, which is considered a common indicator of the development of multiple organ failure [5, 6].

Although vasoactive drugs such as epinephrine and norepinephrine increase the perfusion pressure as well as cardiac output, they are capable of diverting blood flow away from the mesenteric circulation [5, 7], leading to the destruction of the intestinal mucosal barrier, release of inflammatory factors, intestinal epithelial hyperpermeability, bacterial translocation, and multiple organ dysfunction via the circulation [8–10]. Although the underlying mechanism of total organ failure is not entirely understood, it can be avoided, since early diagnosis and management of sepsis have been associated with improved outcomes and reduced mortality [11].

The process of early detection of sepsis may be straightforward; despite this, the early stage of sepsis remains challenging for patients [11]. Routine assessments of splanchnic perfusion such as mucosal perfusion, mesenteric blood flow, mucosal oxygenation, intramucosal pH, gastric and arterial partial pressure of carbon dioxide, intestinal permeability, and lactate production can be used as preventative steps, these tools are not useful once sepsis has already developed [12]. Therefore, there is an urgent need for an accurate, reproducible, radiation-free, rapid, and non-invasive tool to provide information about intestinal perfusion without compromising the treatment of patients with sepsis [13–15].

The use of critical care ultrasound has increased in ICUs in recent years. It has been found to be useful in identifying valuable information to improve medical decision-making, especially in patients with septic shock [16]. Ultrasound can reliably assess arterial and venous blood flow [17, 18]. The SMA is the primary artery supplying the small intestine [19]. In the fasting state, approximately 25% of the cardiac output depends on the splanchnic vascular area. Postprandially, blood flow into SMA is doubled or even tripled [20]. Although effective therapy is still lacking, the therapeutic approach that is selected considerably influences mortality risk. Therefore, in the present study we hypothesized that SMA ultrasound is an effective technique to predict mortality in patients with sepsis.

## Methods

### Study design and patients

This case-control study involved patients with sepsis admitted to the ICU of Qinhuangdao Jungong Hospital from January 2020 to November 2020. The inclusion criteria were as follows: 1) patients meeting the diagnostic Sepsis 3.0 Criteria, with or without the presence of septic shock (within 1 hour of ICU admission); and 2) available SMA ultrasound within the first 24 hours of ICU admission. Sepsis was identified by at least a 2 point increase in the Sepsis-related Organ Failure Assessment (SOFA) score in response to an infection. Septic shock was identified as the need to administer vasopressors for maintaining the average arterial pressure more than or equal to 65 mmHg and the serum lactate concentration more than 2 mmol/L, in spite of sufficient volume resuscitation. Patients who were younger than 18 years, pregnant women, or patients who could not undergo ultrasound (due to abdominal surgery, severe bowel gas, Crohn's disease, ulcerative colitis, or short bowel syndrome) were excluded. The Research

Ethics Committee of Qinhuangdao Jungong Hospital reviewed and approved of this study. All the patients voluntarily participated in the study and provided signed informed consent.

### Data collection and definition

Patients data, including gender, age, APACHE II score, SOFA score, Charlson comorbidity index (CCI), primary infection sites, disease severity (sepsis or septic shock), the use of vaso-pressors within 24 hour after admission, and the baseline laboratory indicators such as platelet count, C-reactive protein, lactate, and albumin levels were acquired from medical records.

SMA ultrasound measurements such as the end-diastolic velocity (EDV), peak systolic velocity (PSV), and resistive index (RI = [PSV—EDV] / PSV) [14, 20] were obtained from the patient records. Two physicians experienced in Doppler ultrasonography performed all evaluations and the reliability test using an ultrasonic system consisting of a 3.5-MHz sector probe (SonoSite, Bothell, WA, USA). The SMA ultrasound parameters were measured within the first 24 hours of ICU admission. The SMA measurement was obtained 1 to 2 cm proximal to the artery [21]. After acquiring a clear image of the right angle of the artery, at least five consecutive cardiac cycles were noted. Three measurements were made, and the final results are presented as the mean of the three readings [22] The prognosis of sepsis was measured by mortality of the patients.

### Statistical analysis

SPSS version 26.0 (SPSS Inc., Chicago, IL, USA), GraphPad Prism version 8.0 (GraphPad, Inc., La Jolla, California, USA), and MedCalc version 18.2.1.0 (MedCalc, Mariakerke, Belgium) soft-ware were used for statistical analysis. The measured variables are expressed as mean ± SD and the categorical values are expressed as number and percentage. The chi-square test, Fisher's exact test, Mann-Whitney $U$ test, or independent Student's $t$ test were used for comparisons between groups. Variables that significantly differed by the univariate analysis ($P < 0.05$) were further analyzed by the multiple logistic regression. The receiver operating characteristic (ROC) curve analysis was used, followed by calculating the area under the curve (AUC), sensi-tivity, and specificity. $P$-value $< 0.05$ was considered to be statistically significant.

### Results

Among 72 patients with sepsis, there were 42 (58.3%) non-survivors and 30 (41.7%) survivors. The mean ages of the survivors and non-survivors were 71.07 ± 7.52 and 71.45 ± 6.89 years, respectively ($P = 0.822$). The mean APACHE II and SOFA scores were higher in the non-sur-vivors than in the survivors (APACHE II, 22.67 ± 4.28 vs. 17.43 ± 3.17 and SOFA score, 10.62 ± 1.67 vs. 7.83 ± 2.29, respectively; $P < 0.001$ for both). The lactate level showed signifi-cant difference between non-survivors and survivors (3.79 ± 1.05 mmol/L and 2.74 ± 0.92 mmol/L, respectively; $P < 0.001$; **Table 1**). The non-survivors showed lower PSV (78.92 ± 4.75 cm/s vs. 100.8 ± 7.10 cm/s), lower EDV (26.09 ± 1.51 cm/s vs. 29.55 ± 1.41 cm/s), and lower RI (0.67 ± 0.03 vs. 0.71 ± 0.02) than the survivors ($P < 0.001$ for all). However, no significant dif-ferences found in the gender, CCI score, original infection sites, occurrence of septic shock, platelet count, C-reactive protein, and albumin level between both the groups. Compared with survivors, non-survivors experienced higher dose of norepinephrine (**Table 1**).

The statistically significant variables included APACHE II score, SOFA score, lactate, low PSV, EDV, and RI which evidenced to be the potential risk factors for mortality by the univari-ate analysis ($P < 0.05$) were used to establish a stepwise multivariate logistic regression model. However, only PSV was included in the model. Patients with higher PSV (100.8 cm/s) have a 70.5% lower chance of dying (OR = 0.295; 95% CI: 0.094–0.925) (survivors) than patients with lower PSV (78.92 cm/s) (Table 2).

**Table 1. Characteristics of septic patients (n = 72).**

| Characteristics | Survivors (n = 30) | Non-survivors (n = 42) | P-value |
|---|---|---|---|
| Age (years) | 71.07±7.52 | 71.45±6.89 | 0.822 |
| Gender | | | 0.612 |
| Female | 11 (36.7%) | 13 (31.0%) | |
| Male | 19 (63.3%) | 29 (69.0%) | |
| APACHE II score | 17.43 ± 3.17 | 22.67 ± 4.28 | <0.001 |
| SOFA score | 7.83 ± 2.29 | 10.62 ± 1.67 | <0.001 |
| CCI score | 2.33 ± 1.24 | 2.95 ± 1.36 | 0.052 |
| Original infection sites, n (%) | | | 0.699 |
| Respiratory system | 24 (57.1%) | 20 (66.7%) | |
| Digestive system | 10 (23.8%) | 6 (20%) | |
| Other sites | 8 (19.1%) | 4 (13.3%) | |
| Septic shock, n (%) | 11 (36.7%) | 24 (57.1%) | 0.087 |
| NEmax (μg/kg/min) | 0.26 ± 0.08 | 0.38 ± 0.08 | <0.001 |
| Platelets (×$10^9$/L) | 156.47 ± 44.14 | 169.19 ± 45.91 | 0.243 |
| C-reactive protein (mg/L) | 91.4 ± 57.21 | 100.98 ± 80.13 | 0.577 |
| Lactate (mmol/L) | 2.74 ± 0.92 | 3.79 ± 1.05 | <0.001 |
| Albumin (g/L) | 30.55 ± 3.34 | 29.09 ± 4.15 | 0.116 |
| Ultrasound data | | | |
| PSV (cm/s) | 100.8 ± 7.10 | 78.92 ± 4.75 | <0.001 |
| EDV (cm/s) | 29.55 ± 1.41 | 26.09 ± 1.51 | <0.001 |
| RI | 0.71 ± 0.02 | 0.67 ± 0.03 | <0.001 |

APACHE II: Acute Physiology and Chronic Health Evaluation II; SOFA: Sequential Organ Failure Assessment Score; CCI: Charlson Comorbidity Index; NEmax: maxium dose of norepinephrine; PSV: Peak systolic velocity; EDV: End diastolic velocity; RI: Resistive index.

The ROC analysis showed that the AUC of PSV (0.99) was the largest among variables of the APACHE II score, SOFA score, lactate level, and EDV. The sensitivity, and specificity of PSV were 0.99, and 0.96, respectively (**Fig 1**).

## Discussion

The present results demonstrate that low PSV was an independent risk factor for mortality, and the results of the ROC curves analyses showed that the PSV value had a good diagnostic

**Table 2. Univariate and multivariate logistic analysis.**

| Characteristics | Univariate logistic analysis | | Multivariate logistic analysis | |
|---|---|---|---|---|
| | Odds Ratio (95% CI) | P-value | Odds Ratio (95% CI) | P-value |
| APACHE II score | 1.449 (1.208,1.738) | < 0.001 | | 0.306 |
| SOFA score | 1.974 (1.433, 2.719) | <0.001 | | 0.104 |
| Lactate (mmol/L) | 2.873 (1.616, 5.108) | <0.001 | | 0.287 |
| Ultrasound data | | | | |
| PSV (cm/s) | 0.295 (0.094, 0.925) | 0.036 | 0.295 (0.094, 0.925) | 0.036 |
| EDV (cm/s) | 0.219 (0.111, 0.432) | <0.001 | | 0.546 |
| RI | <0.001 (0.001, 0.001) | <0.001 | | 0.550 |

APACHE II: Acute Physiology and Chronic Health Evaluation II; SOFA: Sequential Organ Failure Assessment Score; PSV: Peak systolic velocity; EDV: End diastolic velocity; RI: Resistive index; CI, confidence interval

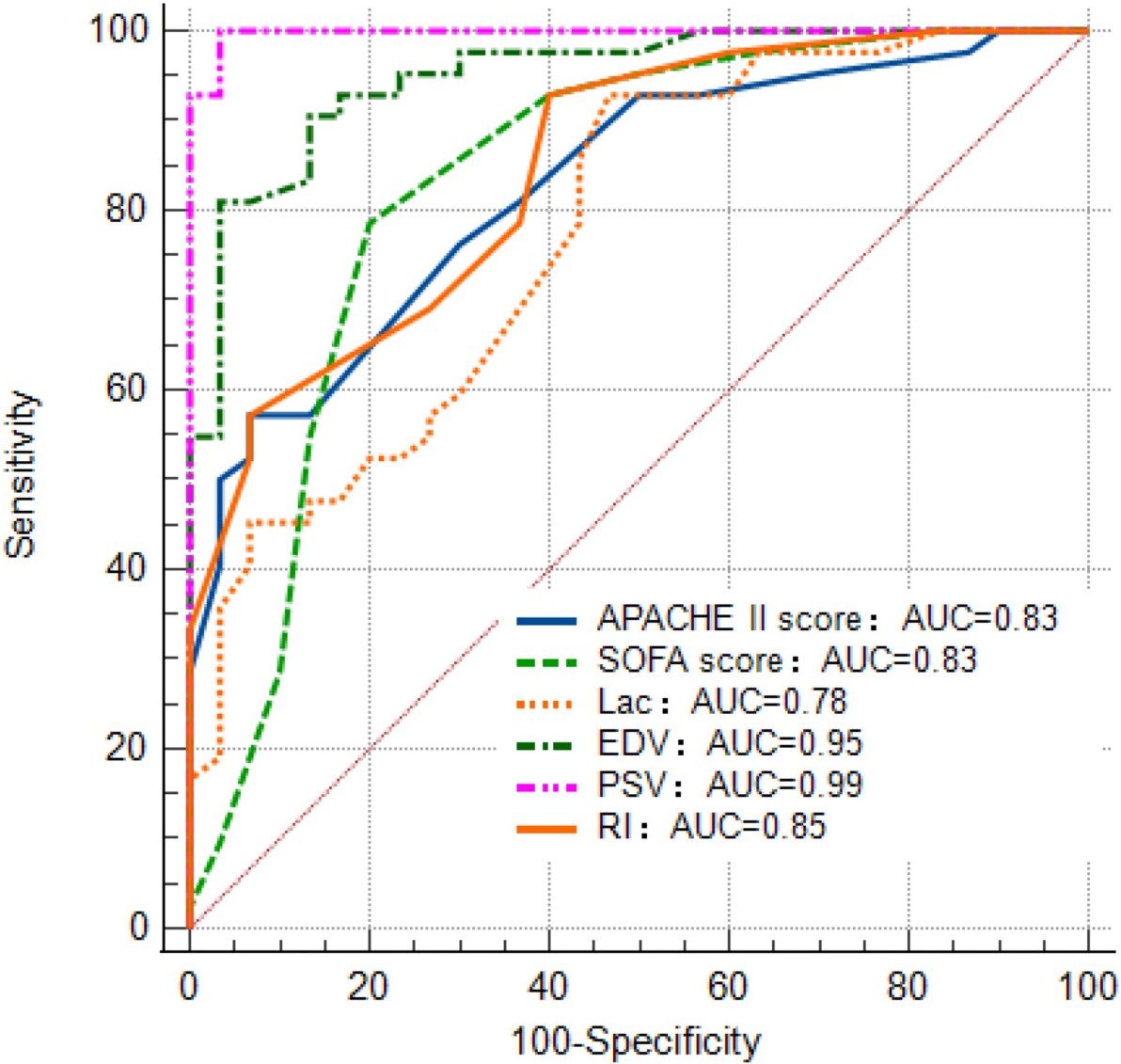

**Fig 1. The area under the receiver operating characteristic (ROC) curve (AUC).** (APACHE II: Acute Physiology and Chronic Health Evaluation II; SOFA: Sepsis-related Organ Failure Assessment; Lac: Lactate; PSV: Peak systolic velocity; EDV: End-diastolic velocity; RI: Resistive index).

performance in predicting mortality in sepsis, which has substantial clinical value for detection in the early stage of sepsis.

In recent years, critical ultrasound has been widely used to evaluate and to aid in the treatment of critically ill patients. Previous studies measured the blood flow velocity in SMA in several pathologies, including patent ductus arteriosus, sepsis, and necrotizing enterocolitis [23]. However, most of these assessed patients with sepsis, allowing for the application of indirect techniques to estimate visceral blood flow, but splanchnic microcirculatory blood flow has rarely been measured [5]. We used SMA ultrasound for the evaluation of patients with sepsis and found that early monitoring of PSV and EDV is extremely valuable in predicting mortality. Although our finding can bridge the knowledge gap, it must be emphasized that our study aimed to conduct a qualitative assessment of the SMA flow, rather than an absolute quantitative comparison between the groups.

The gastrointestinal tract plays a crucial role in the pathogenesis of many diseases, and it is considered to be the motor of multiorgan failure [24]. Non-occlusive mesenteric ischemia is reported to often occur in critically ill patients or in those with significant comorbidities. It is caused by a circulatory failure as well as vasoconstriction in the mesenteric bed [19]. A previous study reported that as a result of non-occlusive mesenteric ischemia, blood flow in SMA reduced to a great extent compared to the cardiac output, and along with blood flow reduction, mesenteric vascular resistance remarkably increased [25]. Therefore, early monitoring of the mesenteric blood flow is of great importance in predicting and evaluating the outcome of severe diseases.

In this study, we observed low PSV, EDV, and RI in the non-survivor group. PSV and EDV alone showed high sensitivity and specificity in predicting the risk of mortality. Therefore, any therapeutic intervention that targets the splanchnic flow may be of great significance in critically ill patients [24]. Other approaches such as vascular catheterization, dilution techniques, and surface mucosal transducers may be more effective for monitoring blood flow; however, they are more expensive and invasive, and expose the patient to the risk of complications and radiation [14, 24]. The SMA ultrasound aids in the analysis of the superior and inferior mesenteric flow and provides several quantifiable parameters, such as pulsatility index, RI, systolic and diastolic velocities, and blood flow volume, to assess the signal from visceral vessels [26].

Platelets play an important role in the coordinated immune response to an infection; however, during a dysregulated host response such as sepsis, platelet activation can cause inflammatory dysfunction and contribute to organ damage [27]. A previous study reported that a sharp drop in the serum albumin level due to bacteremia seriously affected the prognosis [28]. However, no significant differences were observed between the survivors and non-survivors with regard to platelet count and albumin level in our study. Lactate is a reliable prognostic biomarker for poor prognosis in patients with sepsis [12, 28]. However, the mechanism of hyperlactemia is complicated, and several reasons can cause an increase in lactate levels, including tissue hypoperfusion, underlying diseases, drugs, and birth defects. Therefore, the diagnostic value of the lactate level alone may be insufficient [28].

Despite the promising results, our study has some limitations. As only 72 elderly patients were eligible for and analyzed in our study, the study may be prone to sampling bias. This sample size may not be an accurate representation of the target population, and caution must be exercised before extrapolating the results of our study to patients with sepsis of all ages. Despite this limitation, our results are internally valid.

SOFA scores are frequently used to assess organ dysfunction and are closely related to the risk of mortality, with low scores indicating better outcomes [28]. In our study, a strong negative correlation was observed between PSV and the SOFA score; we found that the PSV value was a better predictor of mortality than the SOFA score. Based on our results, SMA ultrasound should be used at an early stage after patients with sepsis are admitted to ICU, with or without septic shock.

## Conclusion

In conclusion, low PSV, evaluated using the SMA ultrasound, as found to be an independent risk factor and a good predictor for mortality in patients with sepsis. This finding is important for performing serial measurements in future prospective SMA ultrasounds.

## Supporting information

**S1 Checklist. *PLOS ONE* clinical studies checklist.**
(DOCX)

**S2 Checklist. STROBE statement—checklist of items that should be included in reports of observational studies.**
(DOCX)

**S1 File.**
(PDF)

## Author Contributions

**Conceptualization:** Xiaowei Gai, Yanan Wang, Dan Gao, Jia Ma, Caijuan Zhang, Qiuyan Wang.

**Data curation:** Yanan Wang, Dan Gao, Jia Ma, Caijuan Zhang.

**Formal analysis:** Xiaowei Gai.

**Funding acquisition:** Xiaowei Gai.

**Investigation:** Caijuan Zhang.

**Resources:** Xiaowei Gai.

**Supervision:** Qiuyan Wang.

**Validation:** Xiaowei Gai, Yanan Wang.

**Writing – original draft:** Xiaowei Gai.

**Writing – review & editing:** Qiuyan Wang.

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
