## [Decision Letter · Decision Letter 0]

25 Jun 2022

PONE-D-22-04824Risk factors for the prognosis of patients with sepsis in intensive care unitsPLOS ONE

Dear Dr. Wang,

Thank you for submitting your manuscript to PLOS ONE. After careful consideration, we feel that it has merit but does not fully meet PLOS ONE’s publication criteria as it currently stands. Therefore, we invite you to submit a revised version of the manuscript that addresses the points raised during the review process.

First of all congratulate you on your manuscript as it has a lot of potential. I would also like to encourage you to carefully address all reviewers comments to further improve the paper prior to resubmission. Please do not hesitate in contacting us if you need any extra-time. 

We look forward to receiving your revised manuscript.

Kind regards,

Monica Cartelle Gestal, PhD

Academic Editor

PLOS ONE

Journal Requirements:

Reviewers' comments:

Reviewer's Responses to Questions

**Comments to the Author**

1. Is the manuscript technically sound, and do the data support the conclusions?

Reviewer #1: Yes

Reviewer #2: Partly

2. Has the statistical analysis been performed appropriately and rigorously? 

Reviewer #1: Yes

Reviewer #2: No

3. Have the authors made all data underlying the findings in their manuscript fully available?

Reviewer #1: Yes

Reviewer #2: Yes

4. Is the manuscript presented in an intelligible fashion and written in standard English?

Reviewer #1: Yes

Reviewer #2: Yes

5. Review Comments to the Author

Reviewer #1: The manuscript in general is great. But there are some observations to improve your article. The main finding of the research is the doppler from superior mesenteric artery and isn´t cited in the title ou abstract, just the "peak systolic velocity" without describe the artery used.

In the abstract, maybe the word "warranted" could be replaced for "needed, necessary or required".

In the method you could explain better the inclusion criteria to control group.

Some terms along the text can be replaced by others more adequate.

Reviewer #2: Dear authors,

This is a very interesting paper, but there are some important points to be clarified:

- In abstract and method: excluded the hospital’s name/identification. Please, Only cite: in a Chinese hospital.

- In method: please, clarify the Case definition and Control definition.

Did you pair the cases and controls? If yes, by which independent variables? So, explain it.

Was the SMR measured by just one person? Was he a doctor? Explain it. If so, it would be more reliable to have the SMR measurement performed by two researchers/physicians and to have performed a reliability test. It may be a limitation and should be cited.

It is necessary to cite the Odds Ratio (OR) effect measure in the method.

- Results: In table 2, please insert the OR values, in the multivariate analysis, of each independent variable and describe by which variable the model was adjusted.

I think the sentence, on page 7, “Only PSV was found to be statistically different between survivors and non-survivors; a low PSV level increased the odds ratio of mortality (OR = 0.295; 95% CI: 0.094 - 0.925; Table 2)”, would be better this way, analyze and decide: “Patients with higher PSV (100.8 cm/s) have a 70.5% lower chance of dying (OR: 0.295; CI= 0.09-0.92) (survivors) than patients with lower PSV (78.92 cm/s).

- Conclusion: On page 10, “In the conclusion, low PSV and EDV values may help identify patients who are at risk of Mortality”. You can not affirm this point, because you did not show the significance of this variable in the adjusted (final) model.

6. PLOS authors have the option to publish the peer review history of their article (what does this mean?). If published, this will include your full peer review and any attached files.

Reviewer #1: **Yes: **Ana Maria Coêlho Holanda

Reviewer #2: No

---

## [Author Response · Author response to Decision Letter 0]

28 Jun 2022

Dear Editors and Reviewers:

On behalf of all authors, I would like to express my sincere thanks for giving us the opportunity to revise our manuscript. We greatly appreciate the editors and reviewers’ positive and constructive comments and suggestions. They are all valuable and very helpful for revising and improving our paper, as well as the important guiding significance to our researches. We have studied comments carefully and have made correction which we hope meet with approval. Revised portion are marked in red in the paper. The main corrections in the paper and the responds to the reviewers’ comments are as flowing:

1. Response to comment: Is the manuscript technically sound, and do the data support the conclusions?

Response: We have revised the manuscript carefully. It describe a technically sound piece of scientific research with data that supports the conclusions.

2. Response to comment: Has the statistical analysis been performed appropriately and rigorously?

Response: We have revised the manuscript and marked in red in the paper.

3. Response to comment: Have the authors made all data underlying the findings in their manuscript fully available?

Response: We have made all data underlying the findings in the manuscript.

4. Response to comment: Is the manuscript presented in an intelligible fashion and written in standard English?

Response: We have repeatedly and carefully corrected the language in the article, striving to be clear and correct.

5. Response to comment: Review Comments to the Author.

Response to Reviewer 1: We are very sorry for our incorrect terms and words. We have revised the manuscript according to the comments and suggestions. 

Response to Reviewer 2: Special thanks to the good comments. We have carefully revised the manuscript. In this case-control study, patients who met the inclusion and exclusion criteria were analyzed. Ultrasound and clinical data were analyzed and compared between non-survivors and survivors. The ROC curve analysis was also performed to determine the best indicator for predicting mortality. Two physicians experienced in Doppler ultrasonography performed all evaluations and the reliability test. The statistically significant variables that evidenced to be the potential risk factors for mortality by the univariate analysis were used to establish a stepwise multivariate logistic regression model. However, only PSV was included in the model. We have made additions and explanations in the results section.

We would like to express our great appreciation to you for your comments on our paper. We looking forward to hearing from you.

Thank you and best regards.

Sincerely,

---

## [Decision Letter · Decision Letter 1]

8 Aug 2022

Risk factors for the prognosis of patients with sepsis in intensive care units

PONE-D-22-04824R1

Dear Dr. Wang,

We’re pleased to inform you that your manuscript has been judged scientifically suitable for publication and will be formally accepted for publication once it meets all outstanding technical requirements.

Kind regards,

Monica Cartelle Gestal, PhD

Academic Editor

PLOS ONE

Additional Editor Comments (optional):

Reviewers' comments:

Reviewer's Responses to Questions

**Comments to the Author**

1. If the authors have adequately addressed your comments raised in a previous round of review and you feel that this manuscript is now acceptable for publication, you may indicate that here to bypass the “Comments to the Author” section, enter your conflict of interest statement in the “Confidential to Editor” section, and submit your "Accept" recommendation.

Reviewer #1: All comments have been addressed

Reviewer #2: All comments have been addressed

2. Is the manuscript technically sound, and do the data support the conclusions?

Reviewer #1: Yes

Reviewer #2: Yes

3. Has the statistical analysis been performed appropriately and rigorously? 

Reviewer #1: Yes

Reviewer #2: Yes

4. Have the authors made all data underlying the findings in their manuscript fully available?

Reviewer #1: Yes

Reviewer #2: Yes

5. Is the manuscript presented in an intelligible fashion and written in standard English?

Reviewer #1: Yes

Reviewer #2: Yes

6. Review Comments to the Author

Reviewer #1: Dear authors, the recomendations made were done properly. I belive the article is ready for publication.

Reviewer #2: The authors made the requested corrections, making the article clearer. So, I think it is possible to accept the paper.

7. PLOS authors have the option to publish the peer review history of their article (what does this mean?). If published, this will include your full peer review and any attached files.

Reviewer #1: No

Reviewer #2: No

---

## [Editor Report · Acceptance letter]

26 Aug 2022

PONE-D-22-04824R1 

Risk factors for the prognosis of patients with sepsis in intensive care units 

Dear Dr. Wang:

I'm pleased to inform you that your manuscript has been deemed suitable for publication in PLOS ONE. Congratulations! Your manuscript is now with our production department. 

Kind regards, 

on behalf of

Dr. Monica Cartelle Gestal 

Academic Editor

PLOS ONE